**PLOS | GENETICS**

# Genetic mapping of fitness determinants across the malaria parasite *Plasmodium falciparum* life cycle

Xue Li[1], Sudhir Kumar[2], Marina McDew-White[1], Meseret Haile[2], Ian H. Cheeseman[1], Scott Emrich[3,4], Katie Button-Simons[3], François Nosten[5,6], Stefan H. I. Kappe[2,7], Michael T. Ferdig[3], Tim J. C. Anderson[1‡]*, Ashley M. Vaughan[2‡]*

**1** Texas Biomedical Research Institute, San Antonio, Texas, United States of America, **2** Center for Global Infectious Disease Research, Seattle Children's Research Institute, Seattle, Washington, United States of America, **3** Eck Institute for Global Health, Department of Biological Sciences, University of Notre Dame, Notre Dame, Indiana, United States of America, **4** Electrical Engineering and Computer Science, University of Tennessee, Knoxville, Tennessee, United States of America, **5** Shoklo Malaria Research Unit, Mahidol-Oxford Tropical Medicine Research Unit, Faculty of Tropical Medicine, Mahidol University, Mae Sot, Thailand, **6** Centre for Tropical Medicine and Global Health, University of Oxford, Oxford, United Kingdom, **7** Department of Global Health, University of Washington, Seattle, Washington, United States of America

☙ These authors contributed equally to this work.
‡ TJCA and AMV also contributed equally to this work.
* tanderso@TxBiomed.org (TJCA); Ashley.Vaughan@seattlechildrens.org (AMV)

**Data Availability Statement:** All raw sequencing data have been submitted to the NABI Sequence Read Archive (SRA, https://www.ncbi.nlm.nih.gov/sra) under the project number of PRJNA524855.

## Abstract

Determining the genetic basis of fitness is central to understanding evolution and transmission of microbial pathogens. In human malaria parasites (*Plasmodium falciparum*), most experimental work on fitness has focused on asexual blood stage parasites, because this stage can be easily cultured, although the transmission of malaria requires both female *Anopheles* mosquitoes and vertebrate hosts. We explore a powerful approach to identify the genetic determinants of parasite fitness across both invertebrate and vertebrate life-cycle stages of *P. falciparum*. This combines experimental genetic crosses using humanized mice, with selective whole genome amplification and pooled sequencing to determine genome-wide allele frequencies and identify genomic regions under selection across multiple lifecycle stages. We applied this approach to genetic crosses between artemisinin resistant (ART-R, *kelch13*-C580Y) and ART-sensitive (ART-S, *kelch13*-WT) parasites, recently isolated from Southeast Asian patients. Two striking results emerge: we observed (i) a strong genome-wide skew (>80%) towards alleles from the ART-R parent in the mosquito stage, that dropped to ~50% in the blood stage as selfed ART-R parasites were selected against; and (ii) repeatable allele specific skews in blood stage parasites with particularly strong selection (selection coefficient (s) ≤ 0.18/asexual cycle) against alleles from the ART-R parent at loci on chromosome 12 containing *MRP2* and chromosome 14 containing *ARPS10*. This approach robustly identifies selected loci and has strong potential for identifying parasite genes that interact with the mosquito vector or compensatory loci involved in drug resistance.

All the processed data (VCF files, tables etc), analysis code, and the scripts for regenerating the figures are available at https://github.com/emilyli0325/BSA_lifecycle/.

**Funding:** This work was funded by National Institutes for Health (https://www.nih.gov) grant P01 AI127338 (to MF) and NIH grant R37 AI048071 (to TJCA). This work was conducted in facilities constructed with support from Research Facilities Improvement Program grant C06 RR013556 from the National Center for Research Resources. SMRU is part of the Mahidol Oxford University Research Unit supported by the Wellcome Trust of Great Britain. The funders had no role in study design, data collection and analysis, decision to publish, or preparation of the manuscript.

**Competing interests:** The authors have declared that no competing interests exist.

## Author summary

Malaria parasites are transmitted through female mosquitoes where gamete fusion and meiosis occurs, and humans where parasites proliferate asexually. Our work represents the first systematic analysis of malaria (*Plasmodium falciparum*) parasite fitness cross the complete life cycle, exploiting our ability to conduct genetic crosses in humanized mice. We use parasites recently isolated from Southeast Asia, the epicenter of the evolution and spread of *P. falciparum* resistance to the front line antimalarial, artemisinin. Our results provide possible insights into additional loci involved in resistance-associated malaria evolution and spread. The approach described here can be directly applied to study multiple selectable traits in the human parasite *P. falciparum*, such as parasite compatibility with different mosquito vectors, resistance to multiple drugs, and tolerance of temperature increase (fever). We also anticipate that this approach will accelerate genetic studies in other recombining parasites and pathogens.

## Introduction

Parasitic organisms frequently use multiple hosts and have several morphologically and transcriptionally distinctive life cycle stages. Within each host, parasites must circumvent immune defenses and navigate to new tissues. There are frequently extreme bottlenecks in parasite numbers during transmission [1], with rapid proliferative growth within hosts, and intense competition between co-infecting parasite genotypes. For example, the life cycle of malaria parasites involves successive infection of two hosts: female *Anopheles* mosquitoes, where gamete fusion, meiosis and recombination occurs, and humans in which parasites travel from the skin, develop in the liver and then proliferate asexually in the blood stream. Ideally, we would like to understand how natural selection operates across the complete life cycle and document the genes subject to selection pressures at each life cycle stage: during erythrocytic growth, gametocyte production, oocyst development in the mosquito midgut, migration of sporozoites to the salivary glands, transmission from the salivary glands, sporozoite survival in the skin, and establishment and parasite growth during liver stage development and exoerythrocytic merozoite release.

Selection can be directly measured by examining changes in allele frequency across these developmental stages. Shifts in allele frequencies in populations of thousands of progeny generated by experimental genetic crosses provide locus-specific readouts of competitive fitness. For example, deep sequencing of bulk populations containing thousands of recombinants identified genes selected under different regimens in yeast and *Caenorhabditis elegans* [2–5]. Bulk segregant analysis (BSA) has also been successfully applied to studies of several different parasitic organisms including coccidia (*Eimeria tenella*) and the human blood fluke *Schistosoma mansoni* [6, 7]. Our work was inspired by an exciting series of papers applying pooled sequencing approaches (termed linkage group selection in the malaria literature) for mapping genes of interest in rodent malaria parasites [8–13].

Most studies of *Plasmodium falciparum* to date focus only on the asexual erythrocytic stages [8, 14–17], because they can be easily cultured *in vitro* in red blood cells, circumventing the need for humans or great apes, the natural hosts for this parasite. Two new research tools now allow us to examine selection across the complete life cycle of *P. falciparum*. First, we can maintain the complete life cycle of *P. falciparum* in a laboratory setting by using humanized mice [18] in place of splenectomized chimpanzees or human volunteers. These mice contain

**Table 1. Sample collection and sequence statistics.**

| Parasite stage | (1) Early Oocyst | (2) Maturing oocyst | (3) Sporozoite | (4) Liver Stage | (5) *In vivo* Blood | (6) *In vitro* Blood |
|---|---|---|---|---|---|---|
| Collecting time[a] | d4 | d10 | d14 | d21 | d21 | d22-52 |
| Sample collected | 48 midguts | 48 midguts | 200 Salivary glands | 60 mg liver | 50ul blood (3.5% parasitaemia) | 50ul blood (1–4% parasitaemia) |
| Total DNA (ng) | 1,397 | 1,337 | 3,675 | 9,359 | 142 | 154–2,576 |
| Total *P. falciparum* genome copies[b] | 7,563 | 866,299 | 4,726,149 | 12,521,577 | 1,246,535 | 19.1M-279.6M |
| *P. falciparum* DNA percent before sWGA | 0.01% | 2% | 3% | 3% | 30% | 100% |
| Sequencing approach[c] | Amplicon | Amplicon, sWGA-WGS | Amplicon, sWGA-WGS | Amplicon, sWGA-WGS | Amplicon, sWGA-WGS, WGS | Amplicon, sWGA-WGS, WGS |
| Copies of *P. falciparum* genome for sWGA | na | $2\times10^5$ | $2\times10^5$ | $2\times10^5$ | $2\times10^5$ | $2\times10^5$ |
| *P. falciparum* DNA percent after sWGA[d] | na | 88.09% | 86.74% | 97.16% | 95.31% | 97.33%-99.57% |

a, Parasite stages and sample collecting times are as shown in Fig 1. Day 0 was defined as the day mosquitoes took a blood meal with gametocytes from two parents.

b, We qualified the parasite genome copy number in the total DNA using qPCR, and translated this into parasite DNA percentage, using $2.48\times10^{-5}$ ng as the weight of the *Plasmodium* genome.

c, For samples with host contamination or small amounts of DNA isolated, we performed selective whole genome amplification (sWGA) before whole-genome sequencing (WGS). We used amplicon sequencing to trace biases in mitochondrial DNA (mtDNA) transmission in those samples. For *in vitro* blood samples, we performed sequencing both before and after sWGA to evaluate the accuracy of allele frequency estimated after sWGA. To obtain sufficient representation of the bulk segregant samples, we used $2\times10^5$ copies of parasite genome as template for each sWGA reaction and 1,000 copies for amplicon sequencing.

d, *P. falciparum* DNA percentage after sWGA was measured as the percent of reads that mapped to the *P. falciparum* 3D7 genome.

human hepatocytes and are therefore able to support liver stage development of *P. falciparum*. Hence, we can stage genetic crosses between different *P. falciparum* parasites, including parasites recently isolated from infected patients, and sample multiple parasite life cycle stages for measurement of allele frequency changes throughout the life cycle. Second, selective whole genome amplification (sWGA) provides a simple and effective way to enrich *Plasmodium* DNA from contaminating host tissues. This is critical because *Plasmodium* DNA constitutes a very small fraction of DNA present in malaria-infected mosquitoes; likewise, *Plasmodium* DNA makes up a very small fraction of DNA extracted from malaria-infected livers (**Table 1**). sWGA uses short 8–12 mer oligonucleotide probes that preferentially bind to the target genome, rather than random hexamers used in normal whole genome amplification. This approach was pioneered by Leichty and Brisson [19], and protocols for sWGA have been successfully developed to amplify and sequence malaria parasite genomes from contaminating host tissues [20–23].

Artemisinin resistance is currently spreading across Southeast Asia [24]. SNPs in the *kelch13* (PF3D7_1343700) locus on chromosome (chr) 13 underlie resistance and greater than 124 independent alleles have been recorded in a dramatic example of a soft select sweep [25, 26]. One particular allele (*kelch13*-C580Y) is currently replacing other resistant alleles and spreading toward fixation in independent transmission foci in western Cambodia/Laos/Vietnam and the Thailand-Myanmar border [27–29]. Several studies have suggested that mutations within loci other than *kelch13* may provide a permissive background for evolution of artemisinin resistance or play a compensatory role [30, 31], but the role of such accessory loci is poorly understood.

In this study, we measured skews in allele frequencies across the genome in the progeny of a genetic cross between artemisinin resistant (ART-R, *kelch13*-C580Y) and ART sensitive

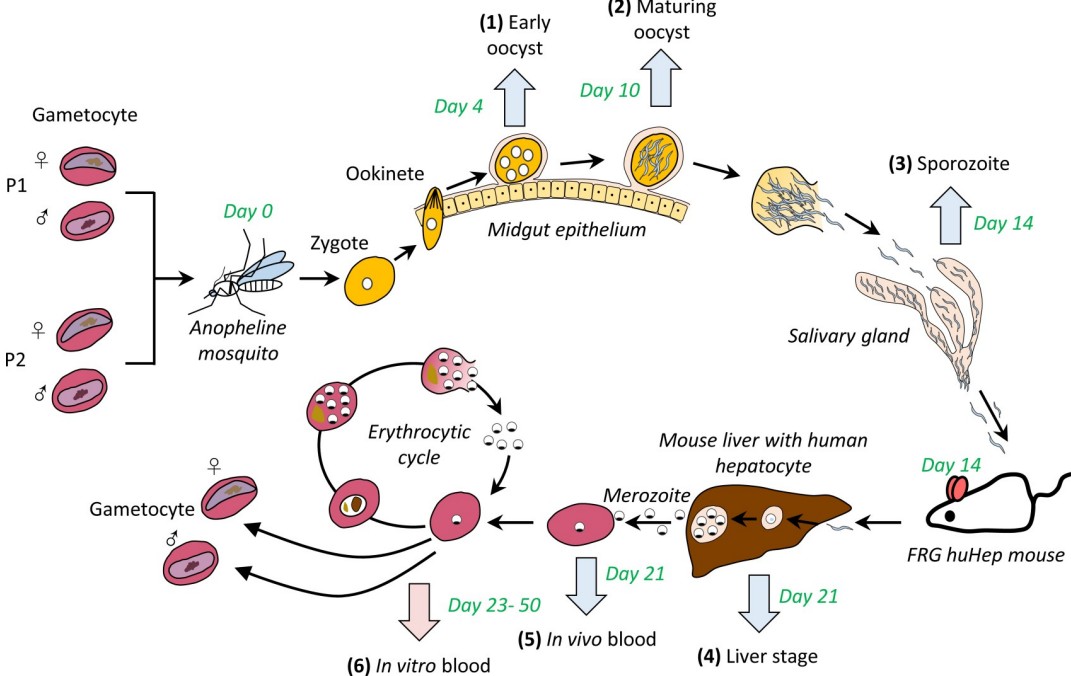

**Fig 1. Genetic mapping of parasite competition throughout the *Plasmodium falciparum* life cycle.** We generated genetic crosses using *Anopheles stephensi* mosquitoes and FRG huHep mice. We collected midgut and salivary glands from infected mosquitoes, infected mouse liver and emerging merozoites from *in vivo* blood, and recovered aliquots of *in vitro* cultured progeny parasites at intervals of 30 days (marked with arrows, parasite stages 1–6). Cross generation and sample collection were completed in two months (marked in green). For samples with host contamination or small amounts of DNA isolated (blue arrows, Table 1), selective whole genome amplification (sWGA) was performed before Illumina whole-genome sequencing (WGA). We used amplicon sequencing to trace biases in mtDNA transmission in those samples. For *in vitro* blood samples (pink arrow), we performed sequencing both before and after sWGA to evaluate the accuracy of allele frequency after sWGA.

(ART-S, *kelch13*-WT) parasites throughout the life cycle to identify genes that influence parasite fitness in parasite stages infecting both the mosquito and vertebrate host. We used ART-R and ART-S parental parasites in order to examine loci contributing to fitness and compensation for deleterious effects of ART-R alleles [16]. We used the humanized mouse model to allow parasite liver stage development of the genetic cross progeny, sWGA to enrich parasite DNA from host contamination and pooled sequencing to determine temporal changes in allele frequency and characterize genomic regions under selection. Our results demonstrate pervasive selection across the parasite genome over the course of a single parasite generation, selection against progeny produced from selfed matings, and strong locus-specific selection against parasite loci on chr 12 and 14.

## Results

### Identification of high-confidence SNPs between parents

*P. falciparum* NHP1337 and MKK2835 were cloned by limiting dilution and used as parents for genetic crosses. Both parasites are from the Thailand-Myanmar border. MKK2835 (ART-S) is a *kelch13* wild-type ART-susceptible parasite collected from a patient who visited the clinic in 2003 prior to the spread of ART resistance [32]. NHP1337 is a recent cloned ART-R parasite, that cleared slowly (Clearance half-life ($T_{1/2}P$) = 7.84 h) from the blood of a patient treated with artemisinin combination therapy in 2011 and carries the C580Y *kelch13*

mutant. Parasites with the C580Y mutation have been rapidly spreading in Southeast Asia and are replacing other ART-resistant *kelch13* alleles [25, 33]. We detected 9,462 high confidence SNPs– 1 SNP per 2.43kb–between the two parental strains from the 21 Mb core genome (defined in [34]) (**S1 Fig**). Comparisons with single clone Southeast Asian parasites from the Sanger pf3k project (ftp://ngs.sanger.ac.uk/production/pf3k/release_5/), shows that the parental parasites both fall into the group designated as KH1 [35] (**S7 Fig**).

## Genetic cross and generation of segregant pools

To generate segregant pools of progeny, we crossed NHP1337 and MKK2835 (**Fig 1**). We fed 500 mosquitoes with a ~50:50 gametocyte mixture of the two parental parasites. Recombinant progeny are generated after gametes fuse to form a diploid zygote that then rapidly transforms into a short-lived tetraploid ookinete which migrates to the basal lamina of the mosquito midgut and transforms into an oocyst. Mitotic division of the 4 meiotic products ultimately leads to the generation of approximately 3000 haploid sporozoites within each oocyst [36]. Oocyst prevalence was 80% with an average burden of three oocysts per mosquito midgut (range: 0–6), giving an estimate of 12 (3×4) recombinant genotypes per mosquito. We dissected a proportion of the infected mosquitoes to collect midguts (48 at each time point) for monitoring allele frequencies during oocyst development. Salivary gland sporozoites from 204 mosquitoes were pooled together and injected in to a single FRG huHep mouse.

We collected samples for allele frequency analysis from infected mosquito midguts, infected mosquito salivary glands, infected humanized mouse livers and infected blood (both mouse blood and injected human red blood cells) after the liver stage-to-blood stage transition. We then recovered aliquots of *in vitro* cultured progeny parasites at two-four day intervals over 30 days (**Fig 1**). We also set up cultures to enrich gametocytes from the *in vitro* cultures. These samples represent the important developmental stages across the parasite life cycle, including early oocyst, maturing oocyst, sporozoites, liver stage schizonts, transitioned blood stage parasites, fifteen asexual cycles in blood stage culture and reproductive gametocytes, required for transmission to the mosquito (**Table 1**). Our experiment examines the impact of selection across one complete *P. falciparum* life cycle.

We measured the total number of parasite genome copies and the amount of host DNA contamination for these segregant pools using qPCR. At the early midgut oocyst stage (4 days after mosquito infection), we isolated ~8,000 copies of the *P. falciparum* genome from 48 mosquito midguts. The parasite DNA represented approximately 0.01% of the total DNA within these isolated midguts. The percentage reached 1.80% after 10 days of mosquito infection, indicating a 196-fold increase of parasite DNA in the six days following initial midgut isolation. The percentage of parasite DNA found in samples from mosquito salivary gland containing sporozoites, liver containing liver stage parasites and liver stage-to-blood stage transitioned *in vivo* blood samples were 3%, 3% and 30%, respectively (**Table 1**).

## sWGA-WGS, WGS and amplicon sequencing

We used three approaches to sequence the segregant pools and quantify allele frequencies: (1) selective whole genome amplification combined with whole genome sequencing (sWGA-WGS), (2) direct whole genome sequencing (WGS) and (3) amplicon sequencing. The methods used were dependent on the level of host contamination and the total amount of DNA present in the samples (**Table 1**). We used multiple methods where possible to determine potential bias. We used the sWGA approach to enrich parasite DNA before WGS in samples with extensive host contamination, including the mosquito and the FRG NOD human-chimeric mouse liver (**Table 1, sample 2–5**). With $0.2×10^6$ copies of parasite genome as template,

the sWGA-WGS approach yielded 0.6–1.4 µg of product after 3h of amplification, of which > 88% was from *P. falciparum*, for both mosquito and mouse samples. By sequencing pools to ~100× coverage, comparable results were obtained between samples prepared by the sWGA-WGS approach and the WGS approach (**Fig 2A, Fig 3 and Fig 4**). We used amplicon sequencing [16] to determine the frequencies of mtDNA from the two parents in those samples for which we used sWGA (**Fig 2C and S2 Fig**). This was necessary because our sWGA primers were specifically designed to minimize amplification of mtDNA, since we were concerned that sWGA with circular DNA would inundate autosomal sWGA products. For day 4 mosquito midgut samples, we only obtained amplicon sequencing data since there was insufficient parasite DNA for a successful sWGA.

## Evaluation of bias in allele frequency measurements

To evaluate the accuracy of allele frequencies estimated after sWGA, we sequenced blood samples using both the sWGA-WGS approach and the WGS approach. We plotted allele frequencies of the parent NHP1337 across the genome and tricube-smoothed the frequency with a window size of 100kb to smooth out noise and estimate changes in adjacent regions. With 10 million 150 bp pair-end sequencing reads, there were fewer loci detected with coverage > 30× by the sWGA-WGS approach relative to WGS (5,024 loci by sWGA-WGS and 7,844 loci by direct WGS). The allele frequency trends, however, were highly consistent after smoothing (**Fig 4A**). The allele frequencies estimated before and after sWGA were strongly concordant ($R^2$ = 0.985, **Fig 4B, S6 Fig**), which strongly supports the comparability of these two different methods.

## Allele frequency changes in segregant pools

a. Mosquito stages: *Plasmodium* sexual blood stage infections differentiate into both male and female gametes and mate; consequently, selfed progeny, resulting from the fusion of gametes from the same parasite genotype, can occur (i.e., NHP1337 male gametes fertilizing NHP1337 female gametes and MKK2835 male gametes fertilizing MKK2835 female gametes). Selection towards selfed progeny is evident from skewing and shifting of whole genome allele frequencies. To investigate population composition at different infection stages, we plotted the allele frequency distribution of *Plasmodium* mitochondria and across the core genome (**Fig 2**). We observed a strong skew (>80%) towards alleles from the ART-R parent in the mosquito stages, which suggests that many selfed progeny from NHP1337 were present.

b. Liver stage: The allele frequency in the progeny parasite population shifted significantly towards the ART-R parent (NHP1337) at the liver stage. This is evident from comparisons of allele frequency distributions in the liver with those from sporozoites (**Fig 2A**, Cohen's *d* test, large effect size = 0.89). This skew observed in the liver stage is reduced in merozoites emerging from the liver (Cohen's *d* test, medium effect size = 0.61).

c. Blood stages: During *in vitro* culture, the allele frequency of NHP1337 (ART-R) dropped to 50%, between day 32 and day 40 (Cohen's *d* test, effect size = 1.65). We maintained replicate *in vitro* blood cultures from day 23 (corresponds to day 2 of *in vitro* blood stage culture). Highly repeatable skews were observed in allele frequencies across the genome in these two parallel cultures (**Fig 2A and 2B**, $R^2$ = 0.985). Furthermore, we observed the same skews in both the mitochondria and across the core genome (**Fig 2C**), strongly suggesting that the selection was against NHP1337 selfed progeny. The NHP1337 selfed progeny were almost eliminated by day 42 and we thus estimated the selection coefficients against the NHP1337

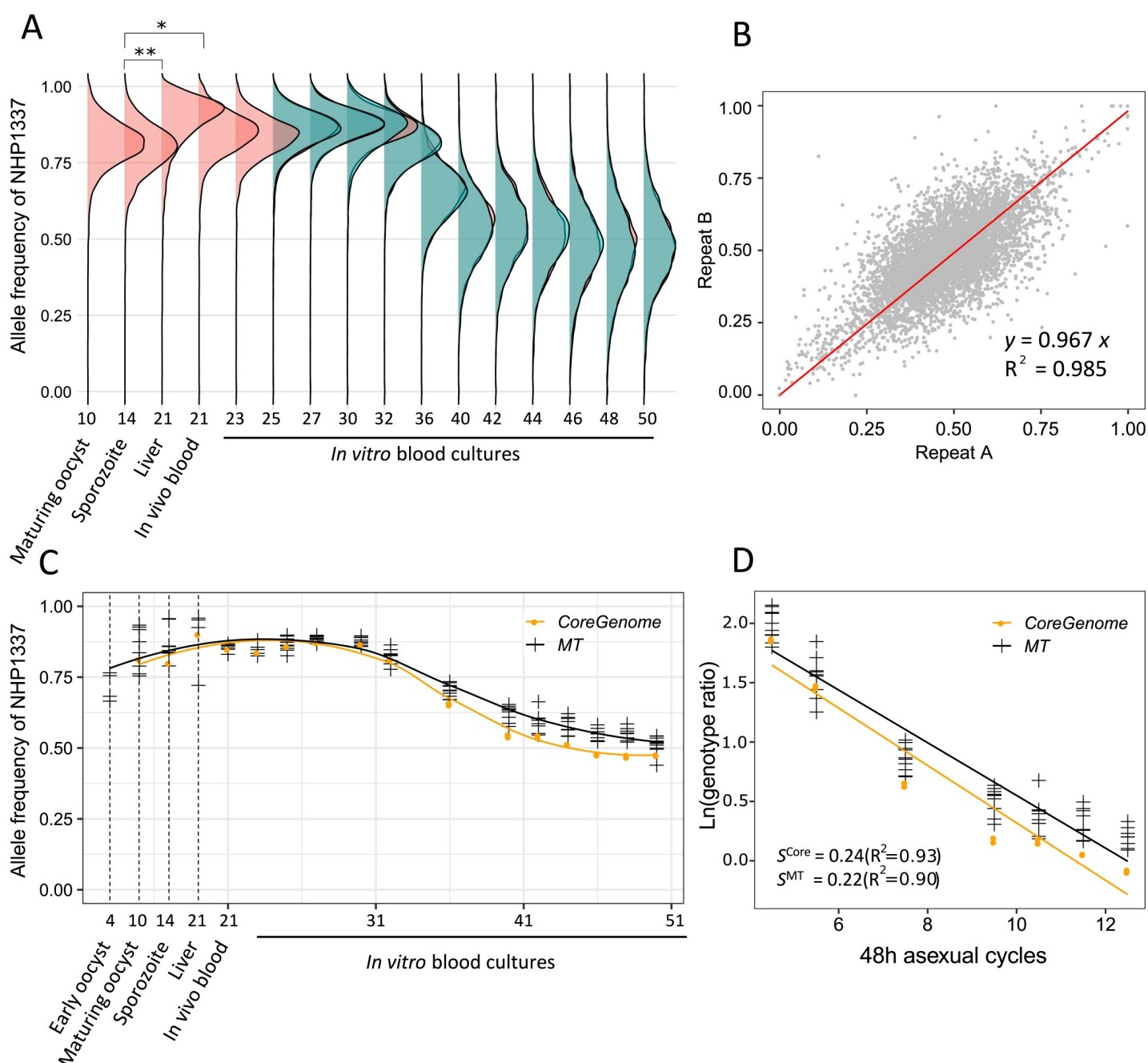

**Fig 2. Change in frequency of the mitochondria and core genome at different infection stages.** (A) Ridgeline plots showed genome-wide allele frequency distributions of NHP1337 throughout the *Plasmodium* life cycle. Each frequency distribution shows the frequency of genome-wide SNPs (9,462) found in progeny bulks at different time points of the parasite life cycle. * indicates Cohen's *d* effect size > 0.5, and ** indicates effect size > 0.8. (B) We detected strong concordance between allele frequencies estimated from experimental replicates. (C) The allele frequency estimated from the mitochondria and core genome showed the same pattern of skew across the life cycle. (D) Natural log of the genotype ratio (NHP1337/MKK2835) plotted against asexual life cycles. The selection coefficient was estimated as the slope of the least-squares fit. Allele frequencies from day30 to day42 were used here. There was no significant difference between fitness costs estimated for the core genome and mitochondria ($P = 0.363$). Positive values of $s$ indicate a selection disadvantage for NHP1337. MT, mitochondria; $s$, selection coefficients; R, correlation coefficient. X-axis in (A) and (C) indicated sample collecting days and corresponding parasite developmental stages.

selfed progeny. We observed strong selection against NHP1337 alleles, with $s = 0.24\pm0.02$ in the core autosomal genome and $s = 0.22\pm0.01$ in mitochondria (**Fig 2D**). There was no significant difference between these two estimates ($p = 0.363$, Least-Squares Means).

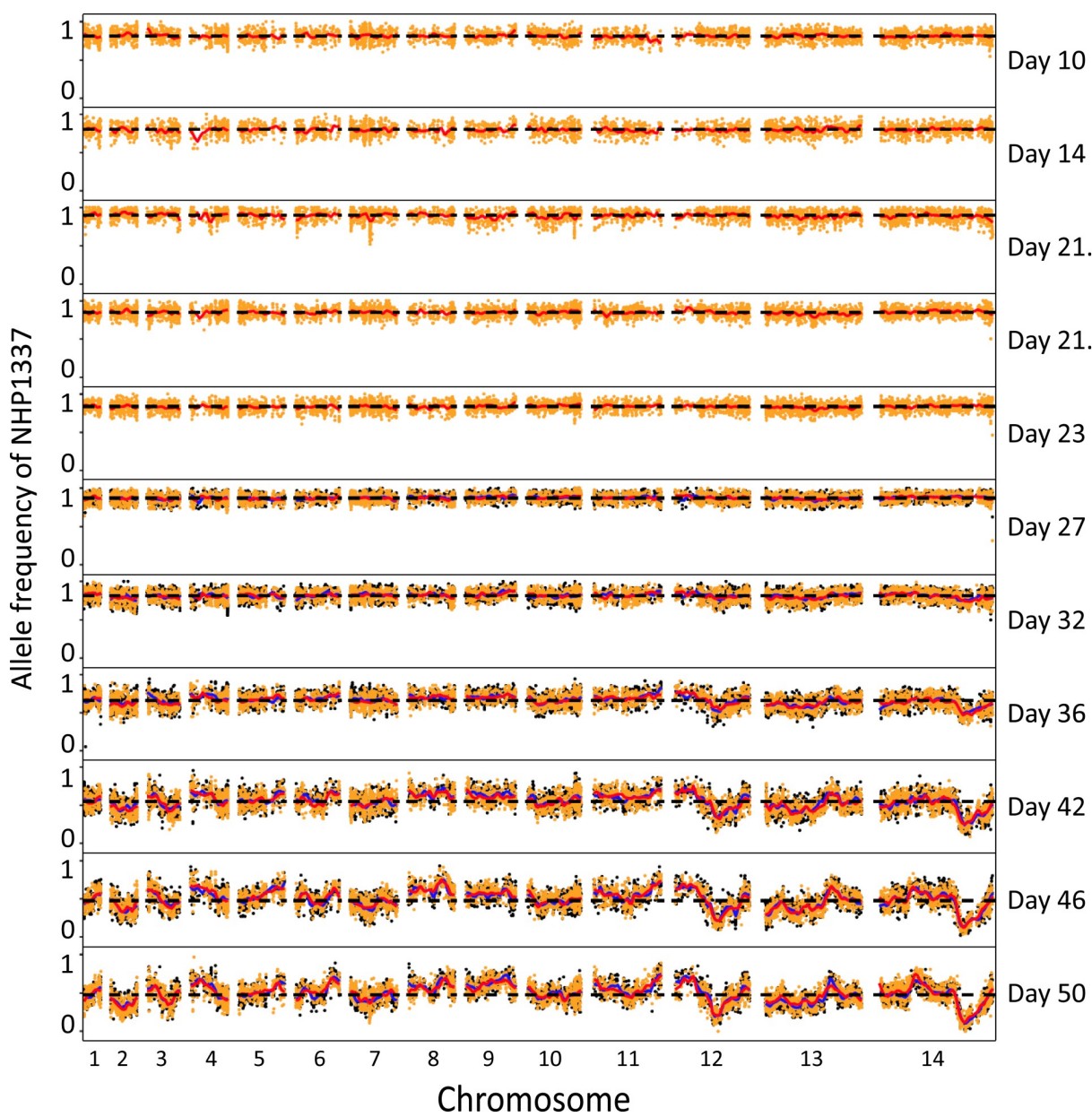

**Fig 3. Plot of allele frequencies across the genome throughout the *Plasmodium falciparum* life cycle.** We divided the parasites into two replicates after two days of *in vitro* culture (day 23). Orange and black indicate allele frequencies from these two parallel cultures. Red and blue lines show tricube-smoothed allele frequencies. Black dashed lines indicate the average allele frequency across the genome. Sample collecting days are marked on the right. Day 10 shows allele frequencies of maturing oocysts, day 14 shows sporozoites, day 21.1 shows liver stage schizonts, day 21.2 shows transitioned blood stage parasites, and day 23–50 shows fifteen asexual cycles in blood stage culture.

d. Gametocyte generation: The sexual commitment (gametocytogenesis) ratio of *Plasmodium* parasites is considered to be generally low (< 3%), but variable among different strains and under different conditions [37–40]. Interestingly, we did not see specific allele frequency changes during the gametocyte enrichment experiment compared to normal *in vitro* blood cultures (**S3 Fig**). These data suggest that progeny from this cross committed to gametocytogenesis at similar rates.

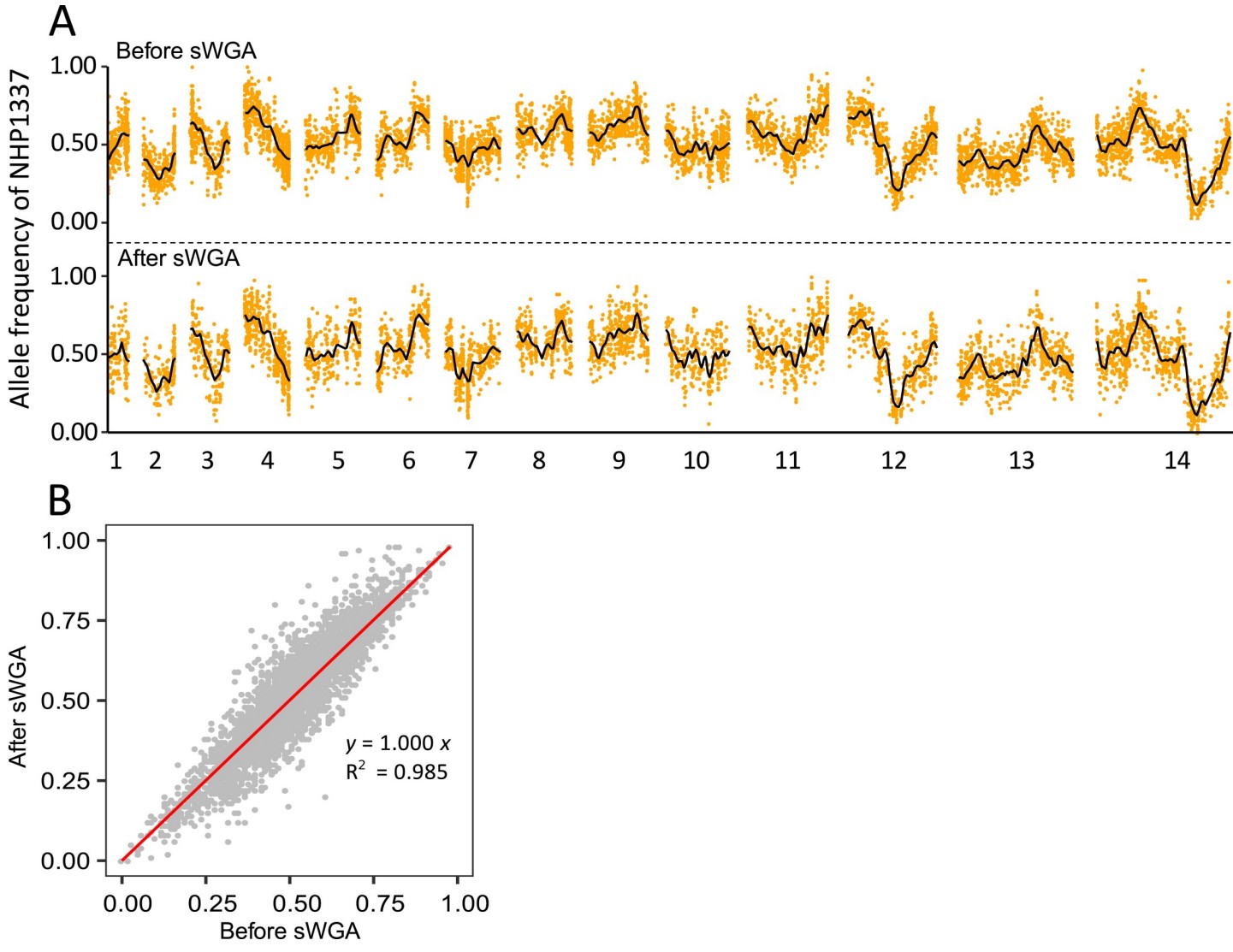

**Fig 4. Allele frequencies estimated before and after selective whole genome amplification (sWGA).** (A) Plot of allele frequencies across the genome. (B) Concordance between allele frequencies estimated before and after sWGA.

## Loci under selection

To pinpoint the loci that determine parasite fitness at each life cycle stage, we first plotted the whole genome allele frequencies throughout the life cycle (**Fig 3**). In addition to the whole genome skew described above, we also observed specific regions of the genome that showed distortion in allele frequency after day 32. The skews in allele frequencies were remarkably consistent between the two replicate blood stage cultures, suggesting pervasive selection at multiple loci across the genome. We calculated G' values to measure the significance of allelic skews (**Fig 5A, S4 Fig and S1 Table**). Two strong quantitative trait loci (QTLs) were identified on chr 12 and 14, with a genome-wide false discovery rate (FDR) < 0.01. We further used Δ (SNP-index) to determine the direction of the allele frequency changes (**Fig 5B and S4 Fig**). In both regions, alleles from NHP1337 (ART-R) were selected against. We then calculated selection coefficients ($s$) across the genome (**Fig 5C**). We observed particularly strong selection at these two QTL regions, with $s = 0.12$ on chr12 and $s = 0.18$ on chr14. In addition, there were a

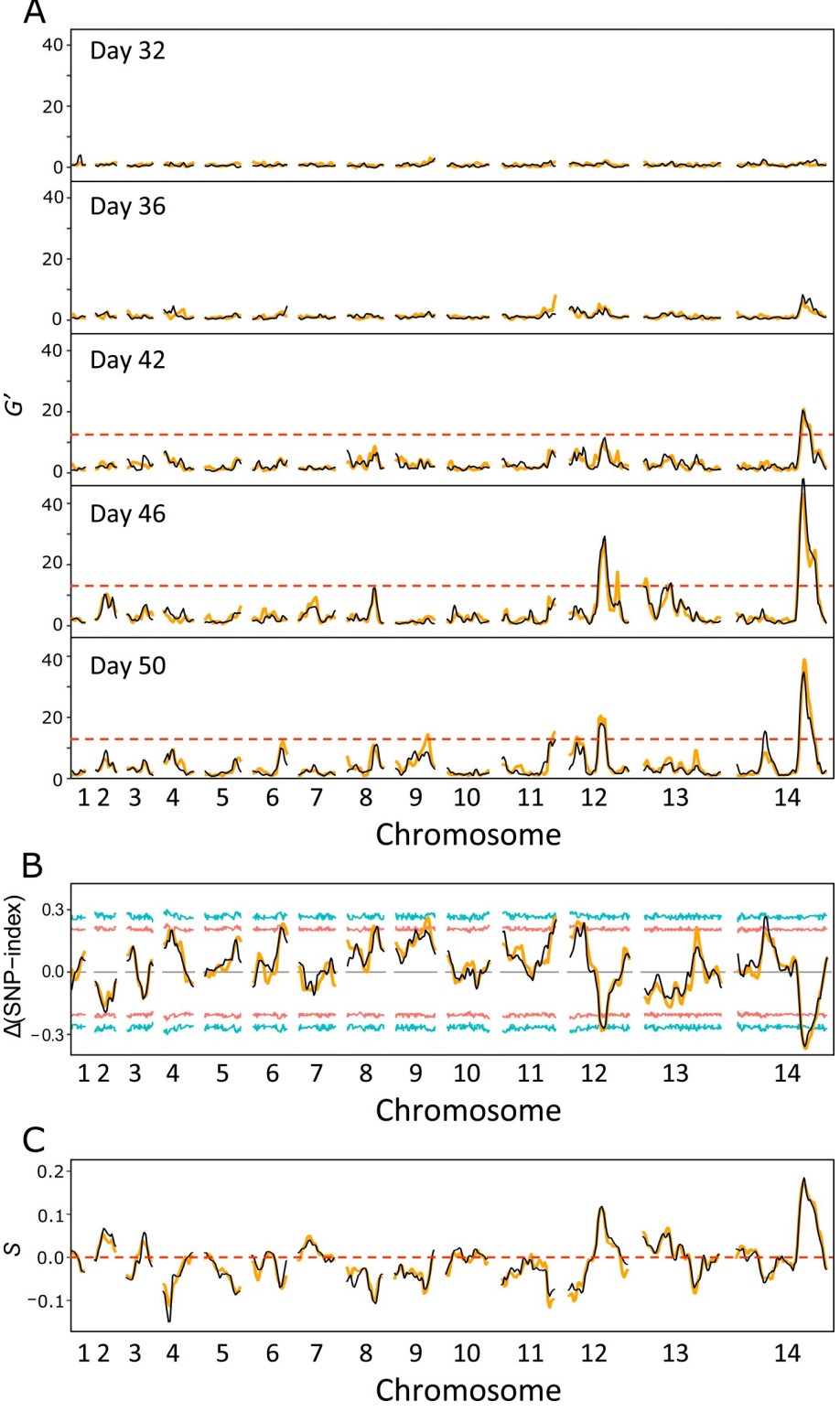

**Fig 5. Bulk segregant analysis.** (A) QTLs were defined with the G' approach by comparing allele frequencies at each locus to the average allele frequency across the genome. Regions with a FDR > 0.01 were considered significant QTLs. (B) Δ(SNP-index) for day50 progeny pools. The Δ(SNP-index) is the difference between the SNP-index of each locus and the genome-wide average SNP-index. A positive Δ(SNP-index) value indicates an increase in alleles from NHP1337. Red and blue lines show the 95% and 99% confidential intervals that match with the relevant window depth

at each SNP. (C) Tricube-smoothed selection coefficients (s). Estimation of *s* was based on the changes of allele frequency from day25 to day50. The mean selection coefficient was adjusted to 0 to remove the influence of selfed progeny. Positive values of *s* indicate a disadvantage for alleles from NHP1337. Orange and black lines indicate experimental replicates.

set of lower confidence QTLs with lower allele frequency changes and less impact on parasite fitness uncovered across the genome (**Fig 5** and **S1 Table**).

## Fine mapping of chr 12 and 14 QTLs

We calculated 95% confidence intervals to narrow down the genes driving selection within the two QTL regions. The QTL on chr 12 ranged from 1,102,148 to 1,327,968 (226 kb) and the QTL on chr 14 ranged from 2,378,002 to 2,541,869 (164 kb).

Chr 12: The QTL region contained 48 genes, with 27 genes bearing at least one non-synonymous mutation differentiating the two parents (**Fig 6** and **S2 Table**). Among the candidate genes with functional annotation, the multidrug resistance-associated protein 2 gene (*mrp2*, PF3D7_1229100) was located at the peak of the chr 12 QTL (**Fig 6A** and **S2 Table**). The *mrp2* allele from NHP1337 carries three indels (3–24 bp) within coding microsatellite sequences compared with that in MKK2835. These indels don't interrupt the open reading frame.

Chr 14: There are 45 genes located in this QTL and 13 contained non-synonymous mutations that distinguish the two parents (**Fig 6** and **S2 Table**). The gene encoding apicoplast ribosomal protein S10 (*arps10*, PF3D7_1460900) was located at the peak of this QTL. There are two non-synonymous mutations (Val127Met and Asp128His) detected in *arps10* from NHP1337 as compared to MKK2835. The Val127Met mutation was suggested to provide a permissive genetic background for artemisinin resistance-associated mutations in *kelch13* in a genome-wide association analysis [31].

## Discussion

### Pervasive selection in a *Plasmodium* genetic cross

In this experiment, we observed both genome-wide selection against selfed progeny, and locus specific selection that resulted in skews in the frequency of particular parental alleles in progeny.

### Genome-wide selection against selfed progeny

Initially, frequencies of alleles derived from the two parental parasites were strongly skewed (0.81 ± 0.08) towards the NHP1337 parent. This deviation from the expected 0.5 ratio for out-crossed progeny occurs because hermaphroditic malaria parasites produce both male and female gametocytes; fusion between male and female gametes of the same genotype (selfing) is possible. The simplest explanation for this observed skew is that an excess of selfed progeny were generated from the NHP1337 parent genotype compared to the MKK2835 parent. We plotted the correlations between mitochondrial frequencies and frequencies of SNPs on different chromosomes across the experiment (**S8 Fig**). Strong correlations show that alleles are not segregated independently, supporting selfing. We also cloned progeny collected on day 23 of the experiment, which confirmed our suspicion that selfing of NHP1337 lead to the skew in allele frequency (Button-Simmons et al. in preparation). Of 212 cloned genotyped progeny recovered, 144 (68%) were ART-R selfed, 5 (2%) were ART-S selfed, and 63 (30%) were recombinant progeny, representing 60 unique recombinants. Interestingly, our bulk sequencing data demonstrated an 86% frequency of alleles from the ART-R parent. In the cloned progeny, we

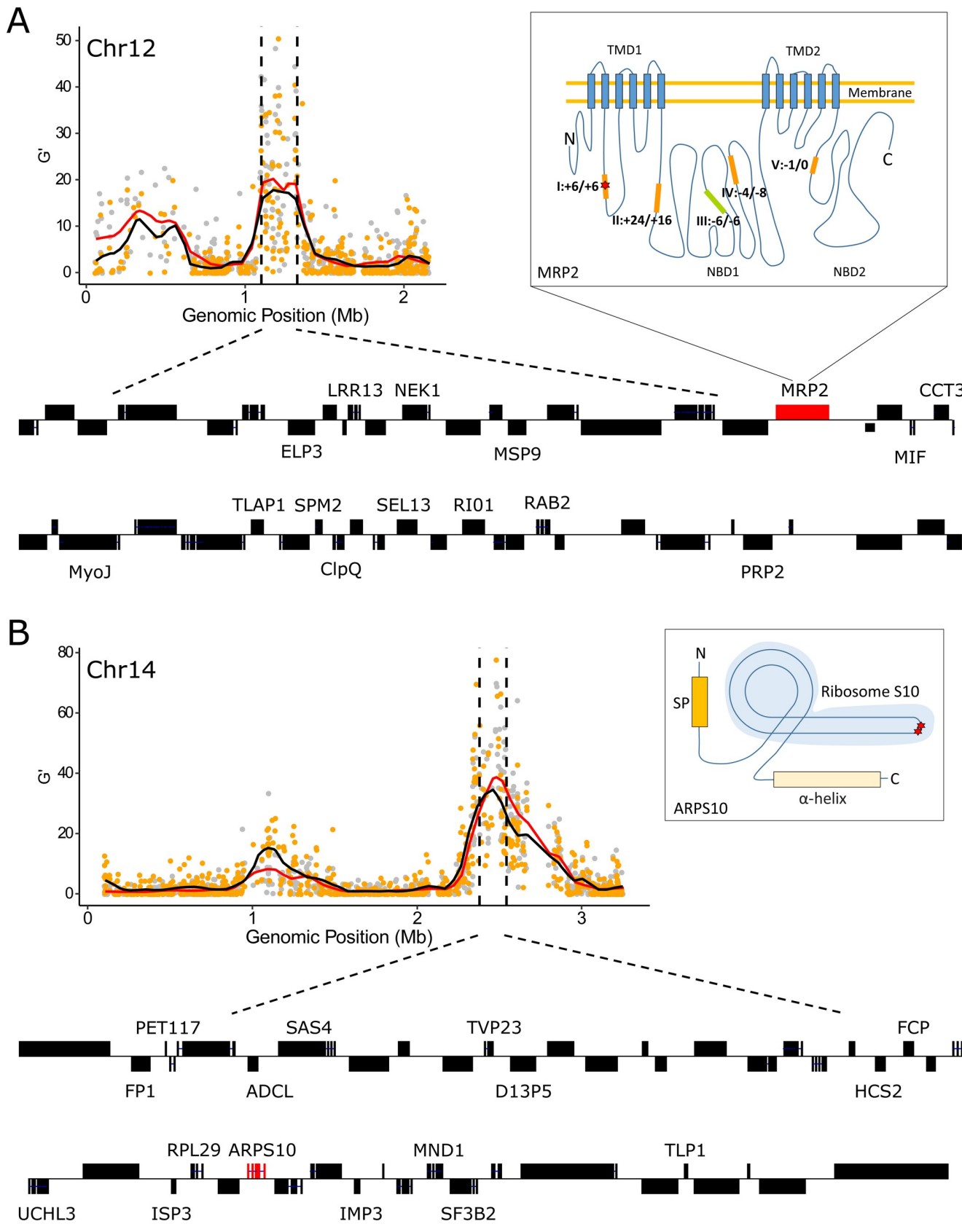

**Fig 6. Overview of the genes inside the QTL regions on chr 12 (A) and chr 14 (B).** Black dashed vertical lines are boundaries of the 95% confidential intervals (CIs) of the QTL. The QTL on chr 12 spanned 226 kb and included 48 genes, and the QTL on ch14 spanned 164 kb and included 45 genes. 2D structure of MRP2 and ARPS10 are presented in boxes next to the G' plot. The structure of MRP2 was adapted from Velga et al., 2014. There are 5 microindels in the coding region of the Pfmrp2 gene (I-V, orange and green blocks). Four of the microindels (orange blocks, 1 SNP and 3 indels) are different between the ART-S and ART-R parental strains. The changes in peptide length relative to *P. falciparum* 3D7 are indicated next to the microindels, as microindel: ART-S/ART-R. ART-S and ART-R parasites have the same amino acid insertion at microindel I, but the sequence includes a synonymous mutation. The structure of ARPS10 was predicted by I-TASSER. ART-R has two non-synonymous mutations in *ARPS10*, Val127Met and Asp128His (red stars). TMD: transmembrane domain; NBD: nucleotide-binding domain; SP: signal peptides.

observed 68% + (30%/2) = 83% extremely close to our estimation from bulk sequencing data. The data from the cloned progeny strongly reinforces our conclusions from the bulk data. It is currently unclear whether the excess selfed progeny from the NHP1337 parent relative to the MKK2835 parent resulted from an imbalance in gametocytes from these parental parasites when staging the cross, or from inherent differences in propensity to self in these two parasite clones.

Frequencies of the NHP1337 genome remained high from day 10 (mature oocysts) until day 30 (after 10 days of *in vitro* blood culture). At this point, genome-wide frequencies of the NHP1337 parasite declined significantly from 0.85 to 0.54 on day 42. We observed a parallel decline of both mitochondrial and autosomal allele frequencies for the NHP1337 parasite. This is consistent with selection removing selfed NHP1337 genotypes from the progeny, otherwise we would expect selection on these two genomes to be decoupled. Selection was extremely strong (mitochondrial $s = 0.22 \pm 0.01$; autosomal $s = 0.24 \pm 0.02$) for both genomes. Furthermore, we observed the same patterns using whole genome sequencing and amplicon sequencing for measuring allele frequencies of mtDNA (**S2 Fig**), suggesting that our results are robust across methodological biases. Analysis of further crosses will allow us to determine whether selection against selfed progeny is a general feature of crosses in malaria parasites.

These data demonstrate systematic selection against genotypes generated by selfing of the ART-R parent. Reproduction by outcrossing is prevalent in nature, even in hermaphroditic species [41]. Inbreeding leads to reduced fitness of offspring (inbreeding depression), while outbreeding among genetically differentiated individuals improves the performance of the F1 generation (heterosis) [41, 42]. We observed strong selection against selfed NHP1337 genotypes which resulted in elimination of selfed progeny in six asexual cycles (day 30–42). Possible explanations for the lower fitness of selfed progeny include: (1) recombination allows removal of deleterious mutations in outcrossed progeny. Accumulation of deleterious mutations occurs during clonal expansion and in inbred parasite lineages. Both parental parasites used in this cross were isolated from Southeast Asia, an area of low parasite transmission intensity, where most infected patients harbor a single parasite genotype. As a consequence mosquito blood meals contain male and female sexual stages from the sample parasite clone, and therefore deleterious mutations can accumulate since inbreeding predominates [43, 44]. We speculate that recombinant genotypes generated by outcrossing between the NHP1337 and MKK2835 parents reduced the numbers of deleterious alleles and therefore outcompeted inbred parental genotypes. (2) *In vitro* culture, where the strongest selection was observed in this experiment, represents an ecological niche change for both parental genotypes. Recombinants generated by outbreeding may be more fit in these laboratory conditions. (3) The selfed NHP1337 parasites that predominate initially are ART-R which may carry a fitness cost relative to ART-S parasites [15, 16, 45]. We note that alleles at the two loci (chr 12 and 14) that are selected against (see discussion section "Locus Specific selection") during days 30–50 are both derived from the ART-R parent.

There is an interesting shift in allele frequencies between sporozoites sampled from mosquito salivary glands and liver stage parasites recovered from infected mice on day 21 (**Fig 2**),

with liver stage parasites carrying high frequencies of NHP1337 alleles (liver 0.89 vs sporozoites 0.79, with large Cohen's *d* effect size [0.89]). The allele frequency of parasites from *in vivo* blood collected on the same day is 0.84, which is between those from sporozoites and liver stage parasites. During liver stage development, single sporozoites take up residency within hepatocytes and divide mitotically over the course of ~7 days (determined with laboratory strains of *P. falciparum* NF54 [46]) until liver schizonts burst releasing tens of thousands of merozoites into the blood. The simplest explanation of the observed allele frequency shift is a genotype-dependent variation in the duration of parasite liver stage development. We suggest that the selfed NHP1337 progeny remain in the liver longer and thus at the day 7 sampling, recombinant liver stage parasites have already transitioned to blood stage, generating the observed difference in allele frequencies. Further work is needed to directly determine the duration of liver stage development and if other liver stage parasite phenotypes (schizont size/ merozoite numbers) differ among parasite genotypes.

## Locus specific selection

We observed a progressive increase in the variance of allele frequencies of SNPs from day 30–50 (during blood stage culture) (**Fig 2A**). Several features of these data suggest that this is primarily driven by selection, rather than genetic drift. First, we noted an extremely strong repeatability in allele frequency skews across the genome in the two replicate parasite cultures established from the humanized mouse infection. This is reflected in the high correlation between allele frequencies between these two replicates at the end of the experiment (**Fig 3**, day 50) when variance in allele frequency is at its maximum. The strong repeatability in patterns of skew observed suggest that there are multiple loci across the genome that influence parasite growth rate and competitive ability. Second, we see several regions of the genome that show extreme skew relative to the genome wide average. Two genome regions in particular (on chr 12 and on chr 14) show strong and significant skews that cannot be explained by drift. These allelic skews also increase progressively from 25–50 days, consistent with selection coefficients (*s*) of 0.18/48 hr asexual cycle for the chr 14 locus and (*s*) of 0.12/48 hr asexual generation for the chr 12 locus.

We observed strong selection against particular alleles segregating in this genetic cross (in the absence of drug pressure). How can such strongly disadvantageous alleles be maintained in natural parasite populations? We suggest three explanations. First, we think that the most likely explanation is that the fitness of these alleles may depend on genetic background [47] and reflect epistatic interactions. We note that of the two parental strains used in this study, MKK2835 (ART-S) was isolated in 2003, while NHP1337 (ART-R), was collected in 2013. In the 10 years between 2003–13, artemisinin-resistant parasites spread to high frequency on the Thailand-Myanmar border [25]. Intense drug selection in this 10-year interval has led to accumulation of additional genetic changes associated with ART-R, which may act epistatically with other ART-R-associated genes [30]. It is certainly interesting that the chr 14 QTL contains *arps10*, which has been suggested to provide a permissive background for ART-R evolution [31]. Outcrossing between individuals with different adaptations can result in disruption of this selective advantage, resulting in a loss of fitness [48]. Further experimental work such as pairwise competition assays between recombinant progeny carrying different chr12 and chr14 haplotypes will be required to confirm the role of these loci, alone or in combination, in determining fitness. Second, there is a possibility that *de novo* deleterious mutations in these two QTL regions were fixed in the cloned NHP1337 parasites during the brief period of laboratory culture. We think this is unlikely because we also see pervasive selection at multiple genes outside these two major QTL regions, just with lower significance using G' statistics. The overall

recombination rate in this cross was 13.8 kb/cM (Button-Simons et al. in preparation). We further counted the recombination events between chr 12 and 14 loci (S5 Table). There were 35 recombination events observed in the 60 unique recombinant progeny between the chr 12 and 14 segments. The recombination between these two loci was even, which indicated that the detection of fitness traits at these two loci were independent. Third, we cannot discount the possibility that the strong selective disadvantages observed within these QTL regions reflects the artificial nature of this system with humanized mice and asexual culturing of parasites. During normal transmission in the field, selection against these genes may not be present.

We note that similar bulk segregant experiments examining fitness determinants in *C. elegans* [5] also detected QTLs with large effect sizes in several different genome regions. The QTL regions identified corresponded to the location of known selfish elements, or co-localized with major eQTLs, consistent with the idea that epistatic interactions are important in fitness related traits. Similarly, analysis of *C. elegans* recombinant inbred lines generated in a 16-parent genetic cross revealed that ~40% of the variance of a key fitness trait (fertility) resulted from epistatic interactions between loci [49]. These *C. elegans* papers support the argument (above) that variation in fitness may be retained within natural populations due to epistasis among the genes involved.

We anticipate that intensity of competition among parasite clones within infected patients may closely parallel the patterns we observed within our genetic cross. The estimated occurrence rate of mixed infections ranges from 18% to 63% in African and Southeast Asia countries [43, 50]. Although there was likely more intense competition in this experimental cross, with millions of sporozoites infecting a single mouse, single cell sequencing has revealed seventeen unique clones in a single human infection [51], which suggests that similar competitive interactions will also occur in patients. We note that while the intensity of competition may be similar in humanized mice, *in vitro* parasite cultures or infected humans, the nature of selection may differ. In infected people, parasite genotypes that allow evasion of immunity or alter parasite cytoadhesion properties may be selected, while growth competition is likely to be the predominant selective force in immunosuppressed humanized mice or *in vitro* culture.

### What drives QTL peaks on chr12 & chr14?

Inspection of the genes under the QTL peaks allows us to speculate about the specific genes that may be driving the selection observed. Miotto et al. (2015) showed that four different non-*kelch13* loci (ferredoxin, *fd*; apicoplast ribosomal protein S10, *arps10*; multidrug resistance protein 2, *mdr2*; chloroquine resistance transporter, *crt*) are associated with the resistance phenotype, but not directly responsible for resistance. They suggested that a suite of background mutations was a prerequisite for mutations in *kelch13*. In our experiment, *arps10* falls near the peak of the strongly selected chr 14 locus (Fig 6), which could suggest a functional relationship. We examined the presence of the background mutations found in both parental strains. The ART-R parent, NHP1337, contains mutations in all four of the genes described by Miotto et al. [31] (*fd*, *mdr2*, *crt* and *arps10*), while the ART-S parent, MKK2835, contains three of these mutations (*fd*, *mdr2* and *crt*), thus only *arps10* mutations are segregating in this cross. It will be interesting to test the role of the remaining three loci (*fd*, *mdr2* and *crt*) by conducting additional experimental crosses.

The multidrug resistance-associated proteins (MRPs), belong to the C-family of ATP binding cassette (ABC) transport proteins that are well known for their role in multidrug resistance. Rodent malaria parasites encode one single MRP protein, whereas *P. falciparum* encodes two: MRP1 and MRP2 [52]. Several studies have shown that PfMRP1 is associated with *P. falciparum's* response to multiple anti-malaria drugs and that disruption of PfMRP1

influences the fitness of parasites under normal culture conditions [53–55]. The function of PfMRP2 is not as well understood. Transfection studies have shown that MRP2-deficient malaria parasites are not able to maintain a successful liver stage infection [52, 56]. In our study, *mrp2* was found located at the peak QTL on chr12. We speculate that *mrp2* may also play a role in parasite fitness during asexual parasite stages. However, we cannot exclude that other neighboring loci may drive the observed allele frequency changes. To confirm the roles of individual genes inside the QTL regions (on both chr12 and chr14), gene-editing studies will be required.

We further analyzed the ancestral/derived allelic state of genes inside the QTL regions of chr 12 and chr14 (S1 File, S6 Table). For both *pfmrp2* and *pfarps10*, the selected alleles in the ART-S parent contained ancestral, rather than derived alleles. Deleterious derived alleles in the ART-R parasites may therefore explain the skews observed on chr12 and 14. We also analyzed the allele frequency distribution among Southeast Asia parasites. We used 678 single clone Southeast Asian parasite lines from the Sanger pf3k project (ftp://ngs.sanger.ac.uk/production/pf3k/release5/). For pf*mrp2*, the selection was against minor alleles, while for *pfarps10*, selection was against major alleles.

## No selection against the *kelch13*-C580Y allele conferring ART resistance

Interestingly, we did not see evidence for selection against the *kelch13*-C580Y allele (chr 13) that underlies resistance to ART treatment. We previously used CRISPR/Cas9 editing to insert the C580Y substitution to a wild type parasite [16]. Head-to-head competition experiments revealed strong fitness costs ($s$ = 0.15/asexual cycle) associated with this substitution. In agreement, Straimer et al [15] conducted similar experiments with Cambodian parasites: they showed that the addition of the C580Y mutation resulted in strong fitness costs for some parasites, but had no fitness impact in recently isolated Cambodian parasites. These data also suggest that epistatic interactions with other loci may compensate and restore parasite fitness. We suspect that this may also be the case in our experiment.

## Technical considerations & caveats

**Maximizing statistical power.** Our statistical power to detect QTLs is limited by the number of recombinants generated. In our experiment, the mouse was infected with sporozoites from 204 mosquitoes carrying on average of three oocsts. Given that each oocyst is expected to contain sporozoites representing up to four different genotypes (i.e. a tetrad), the number of sporozoite genotypes is 204 × 3 × 4 = 2448 in this cross. We can increase the power of these experiments using mosquitoes with higher infection rates. We routinely obtain an average of 10 oocysts/mosquito, so can potentially increase numbers of recombinants by at least three-fold with the same number of mosquitoes. A second advantage of humanized mice over splenectomized chimpanzees as an infection model is that we can easily increase numbers of humanized mice used per cross. By using independent pools of mosquitoes to infect mice, we can multiply the numbers of recombinants generated, while also establishing true biological replicates of each experiment. A third advantage of the humanized mouse system is that we can stage independent crosses with different pairs of ART-R and ART-S parasites to determine if our conclusions are robust.

In this experiment, we found large numbers of inbred progeny generated by mating between male and female gametes of the same genotype. While we were able to use these to document selection against selfed progeny, this reduces the number of recombinant progeny and therefore limits statistical power for locating QTLs. For example, in our cross we estimated that 2448 sporozoite genotypes were initially used to infect the mouse. However, of these only

30% (estimated from dilution cloning of progeny) were recombinants, while the remaining 70% resulted from selfing (Button-Simmons et al. in preparation). Hence the number of independent recombinants used in this cross was no more than $2448 \times 30\% = 734$. A method that maximizes outcrossing would be particularly useful for future crosses. For example, aphidicolin treatment has been successfully used in rodent malaria systems to kill male gametes [57]. In yeast, BSA experiments use parental strains with different mating types to avoid inbreeding [4]. We are not yet able to do this with malaria *P. falciparum* crosses. The dynamics observed in our cross with selection against selfed progeny followed by allele specific selection reflects important differences between *Plasmodium* and yeast systems.

We expect that representation of individual parasite clones will be uneven within progeny pools. Elevated growth of particular "high fitness" clones can generate step-like changes in allele frequency at the recombination points. This has the potential to generate spurious peaks and to confuse the interpretation of BSA experiments [13, 58]. To identify such abrupt allele frequency jumps, we performed a jump-diffusion analysis as described by Abkallo *et al* [13]. This approach identified three allele frequency jump locations in the first experimental repeat of the day 50 population, while no allele frequency jumps were found in the second experimental repeat (**S3 Table** and **S5 Fig**). One of the three jumps is located at the left end of chr 12 QTL, which indicates the possibility of the chr 12 QTL being generated by clonal growth. However, we detected no allele frequency jumps in the vicinity of the chr 14 QTL.

**Combining BSA with cloning recombinant progeny to detect epistasis.** BSA cannot be used to directly examine epistatic interactions, due to the lack of haplotype information. Fortunately, *P. falciparum* has a key advantage over rodent malaria systems because parasites can be grown *in vitro* and cloned by limiting dilution. Hence, BSA can be complemented by cloning progeny from the same genetic cross and directly examining haplotypes carrying different allele combinations. Furthermore, we can use BSA to directly test for interactions between genes. For example, we suspect that interactions between *kelch13* mutations and *arps10* may drive the skew observed at chr 14. This hypothesis can be directly tested by repeating the cross with parasites that have been edited to remove the *kelch13* mutation or candidate *arps10* mutations, to see if the skew on chr 14 disappears.

**sWGA performance.** The sWGA method efficiently enriched *P. falciparum* DNA from infected mosquito and mouse tissues, confirming the performance of this approach for enriching parasite DNA from dried blood spots [20–23]. Our results further show that sWGA does not generate bias in allele frequency measurement (**Fig 4**). However, sWGA does have limitations with highly contaminated samples, such as early infected mosquitoes (four days post infection). DNA extracted from day 4 midguts typically contains > 99.99% mosquito DNA. Only 4.3% of sWGA products from these samples were *Plasmodium* DNA. In contrast, we were able to obtain > 88% of parasite DNA from sWGA, with starting material containing $\geq 1\%$ *P. falciparum* DNA (**Table 1**).

**Potential of BSA for examining selection in the mosquito stage.** We did not observe allele frequency changes during mosquito infections in this experiment. We suggest two reasons for this. First, the *Anopheles stephensi* mosquito used is originally from urban India and widely spread across Southeast Asia, and therefore may show good compatibility with Southeast Asian parasites. Furthermore, this specific mosquito line has been long-term lab adapted, and is highly susceptible to infection with multiple parasite lines. Second, the infection period in mosquitoes in this experiment is relatively short, because we sacrificed all the mosquitos in two weeks. As a consequence, we can only detect very strong selection at this stage. However, hard selection resulting from incompatibility between parasites and mosquitoes should still be possible to detect and map in this system. We note that Molina-Cruz et al [59] were able to determine parasite QTLs for compatibility between *P. falciparum* and mosquitoes using

parasite progeny derived from the original malaria crosses conducted in chimpanzees, providing proof-of-principal that this is possible.

Human malaria can now undergo liver stage development within humanized mice, while blood stage parasites can be grown *in vitro* in culture and cloned. The power of the BSA approach has been clearly demonstrated in rodent malaria, where it has been used to identify the genetic components controlling a broad range of selectable phenotypes, including virulence and immunity, growth rate and drug resistance [8–12]. However, human malaria parasites and rodent malaria parasites are genetically distant and human parasites show numerous unique biological features not found in rodent malaria parasites. Our approach can now be applied to directly study multiple selectable traits in the human parasite *P. falciparum* via genetic crosses. We anticipate that BSA will provide a powerful approach for the study of *P. falciparum* genetics.

# Material and methods

## Ethics approval and consent to participate

The study was performed in strict accordance with the recommendations in the Guide for the Care and Use of Laboratory Animals of the National Institutes of Health (NIH), USA. To this end, the Seattle Children's Research Institute (SCRI) has an Assurance from the Public Health Service (PHS) through the Office of Laboratory Animal Welfare (OLAW) for work approved by its Institutional Animal Care and Use Committee (IACUC). All of the work carried out in this study was specifically reviewed and approved by the SCRI IACUC.

## Preparation of genetic cross and sample collection

We generated the cross using FRG NOD huHep mice [60] with human chimeric livers and *A. stephensi* mosquitoes as described by Vaughan *et al.* [18] (see S1 File for details). We collected samples from infected mosquito midgut and salivary gland, mouse liver and *in vivo* blood, and *in vitro* blood cultures (**Fig 1** and **Table 1**). We fed ~500 mosquitos with mixed gametocytes from each parent at equal ratio. This day was defined as day 0 for sample collecting. Forty-eight midguts were dissected at each oocyst collection time point (day 4 and day 10). The prevalence of infection was analyzed at day 10. Salivary gland were separated to collect sporozoites at day 14 after infection. Sporozoites from 204 mosquitoes were mixed together for infection into the mouse and for isolation of genomic DNA.

Six days after sporozoite injection (day 20), we injected mice intravenously with 400 μL of packed O+ huRBCs. The intravenous injection was repeated the next day (day 21). Four hours after the second huRBC injection, mice were sacrificed and blood was removed by cardiac puncture in order to recover *P. falciparum*–infected huRBCs. The mouse liver was dissected, immediately frozen in liquid nitrogen and then stored at −80°C.

The infected red blood cells were washed, mixed with equal volume of packed huRBCs, and resuspended in complete medium at 2% hematocrit. Two days after culture, the parasites were split equally into two wells (repeat A and repeat B) of a standard six-well plate. About 50 μL of freshly packed huRBCs were added every 2 days to each replicate. To maintain healthy cultures, serial dilutions of parasites were carried out once the parasitemia reached 4%. The cultures were maintained for 30 days in total (day21-day50), and 50ul packed red blood cells (RBCs) were collected and frozen down every 2–4 days. We also set up gametocyte enrichment cultures from day 32 progeny population with daily medium changes but no fresh huRBCs. Samples were collected 8 days (day 40) and 16 days (day 48) later.

## Library preparation and sequencing

We extracted and purified genomic DNA using the Qiagen DNA mini kit, and quantified amounts using Qubit. We performed real-time quantitative PCR (qPCR) reactions to estimate the proportion of parasite genomes in each DNA sample (**S1 File**, **S4 Table**). We used selective whole genome amplification (sWGA) to enrich parasite DNA for samples obtained from infected mosquito and mouse tissues. We used selective whole genome amplification (sWGA, **S1 File**) to enrich parasite DNA for samples obtained from infected mosquito and mouse tissues. sWGA products were further quantified by qPCR (described above) to confirm that the majority of the products were from *Plasmodium*. We constructed next generation sequencing libraries using 50ng DNA or sWGA product following the KAPA HyperPlus Kit protocol with 3-cycle of PCR.

We used amplicon sequencing to trace the biases in mtDNA transmission, as sWGA with circular DNA may swamp out other sWGA products. We use at least 1000 copies of parasite genome as template for each reaction. Illumina adapters and index sequences were added to the PCR primers (**S4 Table**). Equal number of molecules were pooled from each reaction.

All libraries were sequenced to an average coverage of 100x using an Illumina NEXTseq 500 sequencer.

## Genotype calling

We first genotyped the two parental strains. We mapped the whole-genome sequencing reads against the *P. falciparum* 3D7 reference genome (PlasmoDB, release32) using BWA mem (http://bio-bwa.sourceforge.net/) under the default parameters. To reduce false positives due to alignment errors, we excluded the high variable genome regions (subtelomeric repeats, hypervariable regions and centromeres) and only performed genotype calling in the 21 Mb core genome (defined in [34]). The resulting alignments were then converted to SAM format, sorted to BAM format, and deduplicated using picard tools v2.0.1 (http://broadinstitute. github.io/picard/). We used Genome Analysis Toolkit GATK v3.7 (https://software. broadinstitute.org/gatk/) to recalibrate the base quality score based on a set of verified known variants [34].

We called variants for each parent using HaplotypeCaller and then merged using GenotypeGVCFs with default parameters except for sample ploidy 1. We applied filters to the original GATK genotypes using standard filter methods described by McDew-White et al [61]. The recalibrated variant quality scores (VQSR) were calculated by comparing the raw variant distribution with the known and verified *Plasmodium* variant dataset. Loci with VQSR less than 1 were removed from further analysis.

We generated a "mock" genome using GATK *FastaAlternateReferenceMaker* from the genotype of parent NHP1337 (C580Y). The reads from bulk populations obtained at each stage of the lifecycle were mapped to this genome. Only loci with coverage > 30x were used for bulk segregant analysis. We counted reads with genotypes of each parent and calculated allele frequencies at each variable locus. Allele frequencies of NHP1337 were plotted across the genome, and outliers were removed following Hampel's rule [62] with a window size of 100 loci (**Fig 3**).

## Bulk segregant analysis

We performed the BSA analyses using the R package QTLseqr [63]. We first defined extreme-QTLs by looking for regions with false discovery rate (FDR) < 0.01 using the G' approach [64]. We then calculated the Δ(SNP-index) to show the direction of the selection [65]. Once a

QTL was detected, we calculated and approximate 95% confidence interval using Li's method [66] to localize causative genes.

We also measured the fitness cost at each mutation by fitting a linear model between the natural log of the allele ratio (freq[allele1]/freq[allele2]) against time (measured in 48hr parasite asexual cycles). The slope provides a measure of the selection coefficient ($s$) driving each mutation [67]. The raw $s$ values were tricube-smoothed with a window size of 100 kb to remove noise [68, 69]. A positive value of $s$ indicates selection against alleles from the ART-R parent (NHP1337), while a negative value of s indicates selection for NHP1337 alleles.

## Supporting information

**S1 Fig. Number of SNPs between NHP1337 and MKK2835 in 100kb genome windows.** NHP1337 and MKK2835 differ from the core genome sequence of 3D7 (PlasmoDB, release32) by 13,762 and 13,710 SNPs, respectively.
(TIF)

**S2 Fig.** Mitochondrial allele frequencies estimated by amplicon sequencing (red) and whole-genome sequencing (black).
(TIF)

**S3 Fig. Allele frequencies across the genome following gametocyte enrichment of progeny population.** The enrichment was initiated at day 32. We collected samples for sequencing 8 days (day 40) and 16 days (day 48) later. We compared allele frequencies between gametocyte enrichment cultures (marked as "Gametocyte") and normal *in vitro* cultures (marked as "Mixture") which contained both asexual and sexual parasites.
(TIF)

**S4 Fig. Bulk segregant analysis by Δ (SNP-index) and G' values.** (A) Mapping of loci involved in parasite fitness during malaria parasite life cycle with Δ (SNP-index). (B) *G'* values calculated during mosquito, mouse and early blood stages. *G'* values of day 32–50 samples were shown in Fig 5. Day 21.1 was mouse liver and day 21.2 was *in vivo* blood.
(TIF)

**S5 Fig. Sudden changes in allele frequency identified using a jump-diffusion model.** Location of possible allele frequency jumps (see S4 Table for details) detected are marked by vertical lines; Repeat A and B represent results from two parallel *in vitro* blood cultures; QTL regions located at chr 12 and 14 are marked in grey. The chr 12 QTL has an allele frequency jump detected in the day50 sample in one of the two replicates. No jumps we detected close to the chr 14 in either replicate in the temporal samples examined.
(TIF)

**S6 Fig. Concordance between allele frequencies estimated before and after sWGA.**
(TIF)

**S7 Fig. PCA plot using 678 single clone infection samples from Southeast Asia.** The genotype data was obtained from Sanger pf3k project (ftp://ngs.sanger.ac.uk/production/pf3k/release5/). The parent parasites from the cross analyzed in this study fall into KH1 group as defined by Miotto et al [35].
(TIF)

**S8 Fig. Plot of corrections between mitochondrial frequencies and frequencies of SNPs on different chromosomes across the experiment.** The strong correlations for different genome

regions are consistent with selection against inbred parasites.
(TIF)

**S1 Table. Summary of statistics from bulk segregant analyses.**
(XLSX)

**S2 Table. Genes inside of QTL regions.**
(XLSX)

**S3 Table. Sudden changes in allele frequency identified using a jump-diffusion model.**
(XLSX)

**S4 Table. Primers used in this study.**
(XLSX)

**S5 Table. Recombination events within the chromosome 12 and 14 QTL regions.**
(XLSX)

**S6 Table. Ancestral/derived allelic state analysis under chr12 and chr14 QTL regions.**
(XLSX)

**S1 File. Supplemental methods and materials.**
(DOCX)

## Acknowledgments

SMRU is part of the Mahidol Oxford University Research Unit supported by the Wellcome Trust of Great Britain. We thank Dr. Chris Illingworth for advice on implementing the jump-diffusion analysis.

## Author Contributions

**Conceptualization:** Xue Li, Katie Button-Simons, Michael T. Ferdig, Tim J. C. Anderson, Ashley M. Vaughan.

**Data curation:** Xue Li.

**Formal analysis:** Xue Li.

**Funding acquisition:** Michael T. Ferdig, Tim J. C. Anderson.

**Investigation:** Sudhir Kumar, Marina McDew-White, Meseret Haile, François Nosten, Ashley M. Vaughan.

**Methodology:** Xue Li, Sudhir Kumar.

**Project administration:** Katie Button-Simons, Tim J. C. Anderson.

**Resources:** François Nosten, Stefan H. I. Kappe.

**Software:** Xue Li.

**Supervision:** Ian H. Cheeseman, Tim J. C. Anderson, Ashley M. Vaughan.

**Visualization:** Xue Li.

**Writing – original draft:** Xue Li, Sudhir Kumar, Tim J. C. Anderson, Ashley M. Vaughan.

**Writing – review & editing:** Xue Li, Sudhir Kumar, Ian H. Cheeseman, Scott Emrich, Katie Button-Simons, François Nosten, Stefan H. I. Kappe, Michael T. Ferdig, Tim J. C. Anderson, Ashley M. Vaughan.

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
