## [Decision Letter · Decision Letter 0]

22 Aug 2019

Dear Dr Anderson,

Thank you very much for submitting your Research Article entitled 'Genetic mapping of fitness determinants across the malaria parasite Plasmodium falciparum life cycle' to PLOS Genetics. Your manuscript was fully evaluated at the editorial level and by independent peer reviewers. The reviewers appreciated the attention to an important problem, but raised some substantial concerns about the current manuscript. Based on the reviews, we will not be able to accept this version of the manuscript, but we would be willing to review again a much-revised version. We cannot, of course, promise publication at that time.

Two very detailed reviews were obtained from experts in the field. Both provided favorable opinions about the paper and its potential impact on our understanding of the genetics of P. falciparum, including implications for resistance to artemisinin. They also expressed their enthusiasm for the methods that were employed by the authors and the potential power they provide for classical genetic analysis of all the life cycle stages. However, both reviewers requested significant modifications to the manuscript before it would be acceptable for publication. In particular, the reviewers requested that additional information be provided, including greater details regarding the number of progeny obtained, the frequency of recombination, etc. The requested additional information is described in detail in the reviews. Both reviewers also described the value of including information from analysis of individual progeny to provide validation of some of the conclusions. It seems from the current manuscript that this information has been collected and analyzed and that the authors plan to publish it in a separate manuscript. Incorporating some of these data into the current manuscript would address many of the concerns and criticisms expressed by the reviewers and increase the likelihood of acceptance.

If you decide to revise the manuscript for further consideration at PLOS Genetics, please aim to resubmit within the next 60 days, unless it will take extra time to address the concerns of the reviewers, in which case we would appreciate an expected resubmission date by email to plosgenetics@plos.org.

[LINK]

We are sorry that we cannot be more positive about your manuscript at this stage. Please do not hesitate to contact us if you have any concerns or questions.

Yours sincerely,

Kirk W. Deitsch

Guest Editor

PLOS Genetics

Gregory P. Copenhaver

Editor-in-Chief

PLOS Genetics

Reviewer's Responses to Questions

**Comments to the Authors:**

Reviewer #1: In this manuscript, Li, Kumar et al present their analysis of a Plasmodium falciparum genetic cross, achieved using a humanized mouse model pioneered by the senior author Dr. Vaughan (Vaughan et al 2015 Nature Methods). Using two Southeast Asian parasites, they implement a clever combination of whole-genome sequence analysis (WGS), augmented by selective whole genome amplification (sWGA) when necessary to overcome low yields, and complemented by amplicon sequencing, to track changes throughout the life cycle in the genomes of the resulting progeny. The authors provide good evidence that sWGA yields allele frequencies that after smoothing are highly concordant with WGS data.

Their core results are that the artemisinin-resistant (ART-R) parent NHP1337 had mostly selfed in the oocysts that form in the mosquito post mating between the parental gametes, and that later during blood stage development the ART-R parental sequences were selected against in favor of genomic sequences from the ART-sensitive parent MKK2835. Counter selection was most evident in two large segments on chromosomes 12 and 14.

This is an emerging and powerful technology in the field of malaria research and achieving this cross is an important achievement. The genetic methods described herein are also likely to be interesting and informative not only to malaria researchers but to a broad group of researchers studying genetic crosses and looking for methods to examine progeny using bulk segregant analysis (BSA).

The downsides of this work are that ultimately the authors do not provide any experimental evidence to validate candidate loci in these two chromosomes that they suspect might cause a fitness cost. Nor do they provide any information from analysis of independent recombinant progeny that would further support differences in fitness, measured as relative differences in growth rates. There is also no description of how many independent recombinant progeny they actually obtained, or any information on where and how frequently recombination events occurred.

The authors do mention that validation of these candidate genes is required and I think it is reasonable to not expect that for this report. For the progeny however, I do think it’s reasonable to request that the authors provide more information about how many independent recombinant vs selfed progeny were actually obtained, whether this was done at different time points post blood stage culture initiation (on day 21), and at least some presentation of whether recombination events were detected within the chromosome 12 and 14 loci that could further inform mapping of fitness traits. They should also mention whether any fitness studies have been conducted with progeny, and what that shows to date. If those studies have not been performed, they should cite the caveat of not having that information.

Additional elements of my review are detailed below. Some of this is covered above.

The regions on chromosomes 12 and 14 under apparent selective pressure were large: 226 and 164 kb respectively. The chromosome 12 locus contains 48 genes, of which 27 had at least one non-synonymous mutation that distinguished the two parents. The authors highlight mrp2 as a potential cause of this selection. However, mrp2 showed three indels within microsatellite coding sequences and the authors provide no additional experimental evidence that these indels might be causal for any relative differences in fitness. The segment on chromosome 14 contains 45 genes. These include the apicoplast ribosomal protein S10 (arps10) gene that carries two amino acid substitutions in the ART-R parent, of which the Val127Met mutation had earlier been associated with genetic backgrounds on which mutant K13 emerged (Miotto et al 2015 Nature Genetics). No experimental data are presented herein to confirm any contribution of this gene to fitness, and it may well be a background effect as opposed to a causal effect on parasite fitness.

Fig 2: The authors need to provide more detail on what they are showing with their Ridgeline plots in panel A, as not all readers can be expected to know how these are constructed. How many SNPs across the genome did they use to calculate the allele frequencies? What is the spread? Is this a type of confidence interval averaged across all SNPs? Also, say a SNP from the ART-R parent was seen in one of 30 reads, then is that considered a real value or is there a threshold below which a SNP is removed because it might be a sequence artefact?

 Figure 3 clearly shows that the vast majority of the parasites tested throughout the life cycle post meiosis were selfed progeny of the ART-R parent. An increase in representation of the ART-sensitive MKK2835 genome was only apparent in blood stage parasites from about day 42 onwards (i.e. about day 21 of blood stage culture). The figure clearly the dip in chromosomes 12 and 14, presumably reflecting the selective advantage of one or more loci in each segment. One result that is critically missing is how many progeny did they ultimately recover? Did they initiate cloning at different times including late? What were the numbers of independent vs. selfed progeny over time? Were recombination events observed in the chromosome 12 and 14 segments that could help further refine loci that segregate with fitness? Were fitness assays ever conducted with any independent recombinant progeny? If they only cloned early, what were the numbers of recovered selfed vs independent recombinant progeny?

Figure 4: The authors should detail in the legend on which day they performed their comparison. Presumably it was in late blood stage cultures.

Lines 149-50: “fuse to form a zygotes that then rapidly transforms into a short-lived tetraploid ookinetes” – should be “zygote” and “ookinete”

Line 158: The estimate of 2448 recombinants assumes that all sporozoites were the result of outbreeding. That should be clarified as the authors show that substantial selfing (inbreeding) occurred amongst the progeny.

Line 191: The authors state that they used 3 h of amplification, which is explained in the supplemental file. That file lists only conditions of 35-30°C with Phi29 polymerase. I am not familiar with sWGA. Is that the only condition (other than the later 65°C to inactivate the enzymes), with no denaturation/extension steps? It would be helpful to have that method described in more detail in the supplement.

Line 234: When stating day 23, please specify that this corresponds to day 2 of in vitro blood stage culture, to avoid confusion.

S1 Fig shows a fairly small number of SNPs that differ between the parents. It would be helpful to compare each against the reference 3D7 genome, which I assume would show much greater differences. Also, researchers at the Sanger Institute and colleagues have over the years published a number of descriptions of population subgroups in Southeast Asia (e.g. the KH1-6 subgroups, or more recently the KEL subgroups). Can the authors provide more information on the subgroup affiliations of their parents? Would they consider these to be closely related? One informative way to do this would be to show a PCA plot with these two parents compared to other sequenced Southeast Asian genomes, if possible.

Reviewer #2: General comments:

Excellent research. The work clearly required high technical laboratory skill to conduct. Analysis was sound and thourough. The inclusion of both mosquito and mouse stages has picked up some interesting changes in the parental/F1 populations that would not have been observed with fewer developmental stages. The lack of bias in the sWGA is convincing. I find it partcularly interesting that the kelch allele does not appear deleterious in these genetic backrounds. There is mch that is not explained, but in general most explanations would require further BSA's and/or gene manipulation, which I don't think are essential. I do suggest some extra analysis, as below.

Specific comments/issues:

1. The discussion suggested that it was a surpise to see alleles under string selection, given that the alleles come from clinical isolates. I am not suprised by this, as similar BSA experiments in Drosophila, both yeast models and C. elegans show similar large effects sizes. My suspicion is that this is the result of a large panmictic popultation with a simple elction pressure (eg: P14,L374)

2. The strong skew to he ART-R parent in mosquitio stage is unexpected. Any ideas why this occurs? Would you expect this to occur with another ART-R parent, for example?

3. Selection of inbred progeny does seem to be evident (eg: Fig 2C). Ideally, I would like to see some proof within this manuscript. The ultimate would be cloning and some genotyping of clones. Fig2C could be improved: as the mito genome is only one haplotype, ou could easily correlate allele freq between two regions of one chromosome, or between chromosomes. Strong correlations would show that alleles were not segregating independently, hence supporting the selfing.

4. Why do we see non segregating bias in Fig3 up to day 32, and then segregating allele freq changes after that? WHat do the outcrossed segregants appearing so late (this is not what people observe in yeast BSA experiments). Were selfed progeny lost?

5. Selelection for different alleles or haplotypes could be towards new (derived) alleles or biased for ancestral alleles. Some analysis of alleles/haplotypes WRT allele frequencies in SE Asia and ancestral/derived state might reveal some interesting patterns. This could improve the implications of your BSA considerably.

6. Data availablity. Submitting raw sequence data to SRA is the mimimum requirement. But you can easily do more, which will help other researchers and make re-use of you data more common. I find this increases citations and draws in new collaborators. I suggest making ALL processed data files available (in Figshare for example). This could include: VCF files of variants, data use to generate plots, read counts for alelles, and code used for selection.

Small errors/suggestions:

Page 2, line 39-43: Up and downwards skews could be explained more clearly.

P6,L152: reference for ~10,000 sporozoites

I suggest you clarify at very early in the text that this is only one cycle, not multiple mosquito/mouse/mosquito/mouse/ etc.

P7,L179, and in all other places in the ms, including Table 1: Your measures are unlikely to be accurate to TWO decimal places. Please simplify. eg: 3%, 3% and 30% would be more readable and represent your data better.

P9,L243: What type of test used?

P11,L298: sense in "we also dilution cloned progeny"

P18,L459. Different ART-R and ART-S parents would also improve future studies.

PS: I am in favour of embedding figure legends, figures and tables within the text (rather than at the end) when sumitting manuscripts to journals. Placing them at the end makes reviewing moe diffcult, particularly on trains and flights ... Journals never have a problem with embedded figure in my experience.

**Have all data underlying the figures and results presented in the manuscript been provided?**

Reviewer #1: None

Reviewer #2: No: While raw sequence data are available, I am in favour of processed data (VCF files etc) being made available.

PLOS authors have the option to publish the peer review history of their article (what does this mean?). If published, this will include your full peer review and any attached files.

Reviewer #1: No

Reviewer #2: Yes: Daniel Jeffares

---

## [Decision Letter · Decision Letter 1]

1 Oct 2019

Genetic mapping of fitness determinants across the malaria parasite Plasmodium falciparum life cycle

PGENETICS-D-19-01087R1

Dear Dr Anderson,

We are pleased to inform you that your Research Article entitled "Genetic mapping of fitness determinants across the malaria parasite Plasmodium falciparum life cycle" has been provisionally accepted for publication in PLOS Genetics. Congratulations!

Although, as a front-matter piece, your article will be copyedited, we ask that you please be extra careful to ensure that your work is error free; the corresponding author will have one final opportunity to correct any errors and review the copyedited files when our production team is in contact prior to publication. To this end, the corresponding author(s) and co-authors should now review the accepted files. The corresponding author should send these around to any co-authors as needed, and all authors are strongly encouraged to check the files carefully to ensure that the work is accurate, complete, and optimally formatted. Co-authors must contact the corresponding author, not journal staff, with any correction requests.

If you have a press-related query, or would like to know about one way to make your underlying data available (as you will be aware, this is required for publication), please see the end of this email. Please inform journal staff as soon as possible if you are preparing a press release for your article and need a publication date.

Note to LaTeX users only - please carefully review our Latex Guidelines:

http://journals.plos.org/plosgenetics/s/latex

Now that your manuscript has been accepted, please log into EM and update your profile. Go to https://www.editorialmanager.com/pgenetics, log in, and click on the "Update My Information" link at the top of the page. Please update your user information to ensure an efficient production process.

Yours sincerely,

Kirk W. Deitsch

Guest Editor

PLOS Genetics

Gregory P. Copenhaver

Editor-in-Chief

PLOS Genetics

http://journals.plos.org/plosgenetics/

Reviewer's Responses to Questions

Comments to the Authors:

Please note here if the review is uploaded as an attachment.

Reviewer #1: I commend the authors on a well-constructed rebuttal and revised manuscript. My key concerns as Reviewer 1 have now been satisfactorily addressed. My comments below are minor and can be handled editorially. This is an excellent report that will be well appreciated and that raises interesting findings for follow up investigation.

The authors should specify in their Discussion that further work will be required to assess fitness-specific differences between recombinant progeny in order to confirm roles associated with chromosomes 12 and/or 14. Also they should state that future gene-editing studies will be simportant to confirm the role of individual genes in one or both chromosomal segments (on chromosomes 12 and/or 14). These could be woven into the Discussion, for example on lines 517 and 574.

Table S5: please define what is meant by “mix”. Does this mean several different recombinant break points within those sets of progeny? Also, please list the boundaries of the two chromosomal regions in the footnote and the total length of each chromosome to make this more easily interpretable for readers.

In the revised abstract, there is no need to refer twice to P. falciparum as a human malaria parasite.

Reviewer #2: Thank you for addressing all my concerns.

Have all data underlying the figures and results presented in the manuscript been provided?

Large-scale datasets should be made available via a public repository as described in the 

PLOS Genetics

data availability policy, and numerical data that underlies graphs or summary statistics should be provided in spreadsheet form as supporting information.

Reviewer #1: Yes

Reviewer #2: Yes

PLOS authors have the option to publish the peer review history of their article (what does this mean?). If published, this will include your full peer review and any attached files.

Do you want your identity to be public for this peer review?

 For information about this choice, including consent withdrawal, please see our Privacy Policy.

Reviewer #1: No

Reviewer #2: No

DATA DEPOSITION

If you have submitted a Research Article or Front Matter that has associated data that are not suitable for deposition in a subject-specific public repository (such as GenBank or ArrayExpress), one way to make that data available is to deposit it in the Dryad Digital Repository, http://www.datadryad.org. As you may recall, we ask all authors to agree to make data available; this is one way to achieve that. Please note that Dryad introduced a data publishing charge from 1st September 2013.

The link below will take you to the Dryad record for your article, so you won't have to re‐enter its bibliographic information, and can upload your files directly. More information about depositing data in Dryad is available at http://www.datadryad.org/depositing.

Full information on how to submit your data can be found at the Dryad web site. If you experience any difficulties in submitting your data, please contact help@datadryad.org for support.

http://datadryad.org/submit?journalID=pgenetics&manu=PGENETICS-D-19-01087R1

For more information on PLOS submissions and Dryad, including how to cite your data in your PLOS submission, please visit http://www.plosgenetics.org/static/dryad.action.

PRESS QUERIES

If you or your institution will be preparing press materials for this manuscript, or if you need to know your paper's publication date for media purposes, please inform the journal staff as soon as possible so that your submission can be scheduled accordingly. Your manuscript will remain under a strict press embargo until the publication date and time. PLOS Genetics may also choose to issue a press release for your article. If there's anything the journal should know or you'd like more information, please get in touch via plosgenetics@plos.org.

FMPGENETICS

---

## [Editor Report · Acceptance letter]

7 Oct 2019

PGENETICS-D-19-01087R1 

Genetic mapping of fitness determinants across the malaria parasite Plasmodium falciparum life cycle 

Dear Dr Anderson, 

We are pleased to inform you that your manuscript entitled "Genetic mapping of fitness determinants across the malaria parasite Plasmodium falciparum life cycle" has been formally accepted for publication in PLOS Genetics! Your manuscript is now with our production department and you will be notified of the publication date in due course.

With kind regards,

Matt Lyles

PLOS Genetics

On behalf of:
